# Validation of Angle Estimation Based on Body Tracking Data from RGB-D and RGB Cameras for Biomechanical Assessment

**DOI:** 10.3390/s23010003

**Published:** 2022-12-20

**Authors:** Thiago Buarque de Gusmão Lafayette, Victor Hugo de Lima Kunst, Pedro Vanderlei de Sousa Melo, Paulo de Oliveira Guedes, João Marcelo Xavier Natário Teixeira, Cínthia Rodrigues de Vasconcelos, Veronica Teichrieb, Alana Elza Fontes da Gama

**Affiliations:** 1Departamento de Engenharia Biomédica, Universidade Federal de Pernambuco (UFPE), Recife 50670-901, Brazil; 2Voxar Labs, Universidade Federal de Pernambuco (UFPE), Recife 50670-901, Brazil; 3Departamento de Fisiotarapia, Universidade Federal de Pernambuco (UFPE), Recife 50670-901, Brazil

**Keywords:** RGB-D, biomechanical analysis, Kinect V2, MediaPipe

## Abstract

Motion analysis is an area with several applications for health, sports, and entertainment. The high cost of state-of-the-art equipment in the health field makes it unfeasible to apply this technique in the clinics’ routines. In this vein, RGB-D and RGB equipment, which have joint tracking tools, are tested with portable and low-cost solutions to enable computational motion analysis. The recent release of Google MediaPipe, a joint inference tracking technique that uses conventional RGB cameras, can be considered a milestone due to its ability to estimate depth coordinates in planar images. In light of this, this work aims to evaluate the measurement of angular variation from RGB-D and RGB sensor data against the Qualisys Tracking Manager gold standard. A total of 60 recordings were performed for each upper and lower limb movement in two different position configurations concerning the sensors. Google’s MediaPipe usage obtained close results compared to Kinect V2 sensor in the inherent aspects of absolute error, RMS, and correlation to the gold standard, presenting lower dispersion values and error metrics, which is more positive. In the comparison with equipment commonly used in physical evaluations, MediaPipe had an error within the error range of short- and long-arm goniometers.

## 1. Introduction

Movement evaluation is a handy tool to help treat patients with motor impairment, such as its use by athletes who seek to improve their techniques [1]. Between them, cinematic analysis through motion capture (MoCap) systems are widely used [2]. There are several MoCap types and models in the market, which can be differentiated by the physical principle used: mechanical, magnetic, electronic, and optical. Among these models, the optical MoCap systems of reflective markers stand out and are considered the gold standard for biomechanical evaluation due to their high spatial metric accuracy [3]. In addition, MoCap systems capture patients’ data throughout the analyzed movement, such as joint amplitude, comparison of execution to the biomechanical pattern, posture, and body compensations. For these purposes, some criteria are fundamental, such as the space estimation of the joints and kinematic variables (space, velocity, and acceleration) [4].

In this view, low-cost alternatives represent a viable solution to apply MoCap methods in clinics and laboratories with lower purchasing power. Among the equipment, alternatives are RGB-D (red–green–blue–depth) cameras [5]. These devices include Microsoft Kinect, Intel RealSense, Orbbec Astra, and OpenCV OAK-D, which have a better cost–benefit ratio, are portable, and do not require the use of markers, making their use more practical and convenient for the patient. However, despite the inherent economic advantages, these tools present a trade-off concerning accuracy and precision.

Furthermore, the development of image processing for tracking and machine learning enabled the acquisition of depth estimation with only RGB (red–green–blue) color images [6]. Besides that, algorithms such as OpenPose, MediaPipe, AlphaPose, Detectron2, etc., allow the identification of the human body and the estimation of the position of joints in a three-dimensional (3D) space [7]. Thus, methods that use these technologies can be even cheaper and more practical than RGB-D devices since this technique uses only conventional RGB images provided by any camera.

In general, it is perceived that RGB-D and RGB capture devices present themselves as a new perspective for biomechanical evaluation due to their advantageous characteristics regarding cost and portability. Therefore, a greater understanding of the possibility of application and interchangeability of the devices and data generated is necessary. In addition, the pandemic context has highlighted the need for telemedicine and remote monitoring advances. Based on this need, this work aims to perform a comparative analysis and evaluation of the angular variation of the major limb joints between different RGB-D devices, comparing them with each other together with an RGB solution, having as a confidence parameter a MoCap of reflective markers. This analysis will enable us to understand how far these measurements can be used as a clinical device for rehabilitation purposes.

## 2. Pose Estimation/Recognition Solutions

Human pose estimation and recognition have been used in various applications, from physical therapy in clinics to industrial posture adjustment. For this purpose, different image processing algorithms use CNNs to identify key points. Some of the available technologies are YOLO [8], OpenPose [7], and MediaPipe [9], the last being the one used in this work and described below.

MediaPipe offers a cross-platform open-source framework (e.g., compatible with Android, iOS, web, desktop, cloud, and IoT) developed by Google to build inference streams on multimodal data (e.g., video, audio, time series data) [9]. MediaPipe Pose is a machine learning solution for high-fidelity body pose tracking, inferring 33 3D landmarks across the body from RGB video frames.

MediaPipe creates an insight pipeline that is built as a graph, composed of modular components such as inference models, TFLite. The main use of MediaPipe is to design prototype applications that use joint inference models and other reusable components.

This application performs capture from different perspectives since adding some processing steps and inference components is necessary. Moreover, developing the same application for various platforms can be even more difficult, considering the need to optimize some efforts and alter the processing to suit the target device. To deal with this, MediaPipe creates flows to make transparent and link the different inference models, which are already optimized. As a result, the same pipeline can be used on multiple platforms with the same behavior, allowing the programmer to generate an application on a desktop and deploy it to a cell phone. A clear advantage of MediaPipe is its capability of running on mainstream cellphones (due to its low computational requirements), enabling body-tracking applications to develop these platforms. As a drawback of 3D estimation from 2D content, the Z coordinates of the joints are calculated in relation to a central point of the user instead of having an absolute value, which happens with conventional RGB-D sensors that provide the user’s distance to the sensor.

Pose estimation for health and wellness applications (e.g., gym, dance, etc.) is particularly challenging because of the wide variety of possible poses, degrees of freedom, occlusions, and a variety of body features. BlazePose [10] uses two steps: a detector and a tracker for pose estimation. The first, BlazePose Detector, is a detector that locates the pose region of interest (ROI) within the image. The second, BlazePose GHUM 3D, is used to infer 33 pose reference points across the body in this ROI.

## 3. Material and Methods

This was a transversal observational study where joint angle from monoplanar biomechanical shoulder, elbow, hip, and knee movements were captured simultaneously from three depth sensors, one RGB tracking, and a reflexive marker MoCap System, the last being the reference value. Figure 1 shows the method’s flowchart. The first step is to prepare the volunteer with the attachment of the reflex markers and position them in front of the sensors for simultaneous recording.

With the recorded data, each sensor was preprocessed before the angle calculation. As seen in Figure 1, some sensors needed to undergo a skeleton extraction, identified by the pink and orange flowcharts. By skeleton extraction, we mean extracting the 3D positions of the human body joints. One of the sensors was used as the source of the recordings for the RGB method. After obtaining all the skeletons, the angular variation’s values were calculated, finishing with the filtering and synchronization of the data for the last comparative analyses.

Additionally in Figure 1, each step of the method is separated and identified, starting with the setup containing the equipment, the volunteers, the markers’ positioning, and the volunteers’ recording. In sequence, the sensors are determined by different colors due to the flow each data will follow. The pink color for the gold standard and orange for the equipment without SDK all went through a preprocessing stage to determine the skeleton. The blue flow of the sensor with SDK with this skeleton estimatebranches into the green, representing the copy of the recording with input to the RGB method. All data come together in the angular calculation and pass through a set of filters. The data from the sensors (in green) pass to the interpolation and then synchronize.

### 3.1. Subjects

We developed an experimental study at LACAF (Kinesiology and Functional Assessment Laboratory) of the Universidade Federal de Pernambuco (UFPE). Inclusion criteria were young adults over 18 years old and students at UFPE. The only exclusion criterion for the selection of subjects was the existence of any disease or injury affecting their coordination and range of motion, previously assessed by a physical therapy researcher. No restrictions were imposed on age, weight, height, and gender. Participants were recruited at the end of December 2021. The volunteers are students from the physical therapy department of UFPE. A total of 6 healthy subjects participated of the study: 5 males and 1 female. The associated demographic data are presented in Table 1. The Ethics Committee of UFPE approved the experiment, which was approved under opinion: 3.225.381 (CAAE:03508918.9.0000.5208).

### 3.2. Materials and Setup

This research selected Microsoft Kinect v2, Orbbec Astra, and Intel RealSense (D415 model) as RGB-D capture devices. The Google MediaPipe API was chosen for the RGB application due to its differential in estimating depth with only one RGB camera, besides being a new framework and with few articles for its application in joint tracking. The reference adopted as the gold standard for comparison of angular measurements in this work was obtained using the Qualisys motion capture system.

Three MoCap RGB-D systems were used: Microsoft Kinect v2, Intel RealSense (D415 model), and Orbbec Astra, whose technical specifications are described in Table 2. In addition, the Google MediaPipe framework was used to track and estimate joints in RGB images.

It is important to note that we did not include Kinect Azure in this comparison, which is the newest version of the Kinect sensor. Despite the fact that Kinect Azure provides better accuracy regarding joint and depth estimation compared to its predecessors [11,12], this sensor is not yet globally available; its sale by Microsoft is still restricted to some specific countries. In addition, it was unavailable in our laboratory when this research was developed. For this reason, we used Kinect V2 as a reference as it is estimated for body joints, and depth information is still widely used by the scientific community, even if its production was discontinued by Microsoft.

The captured equipment was arranged in front of the collection area, positioned next to each other, as shown in Figure 2, at a ground elevation of about 1.20 m. The MoCap used was the Qualisys gold standard, equipment with 6 cameras was used as a reference parameter for the angular variation. The recordings were simultaneous for all equipment. However, the recording frequencies differed between the RGB-D/RGB equipment (30 Hz) and the gold standard (180 Hz).

Furthermore, to prepare the volunteers for the Qualisys capture, 24 reflective markers were used in two main configurations (for the upper and lower limbs). For the positioning of the markers, we followed the guide available on the C-Motion Wiki website:Lower Limbs:-Coda Pelvis.-Shank 4.-Thigh 3.-One Segment Feet.Upper Limbs:
-Thorax/Abdomen Model 2.-Upper Arm and Lower Arm Segment Model 1.

### 3.3. Experimental Protocol

The recordings occurred at different times for the lower and upper limbs to facilitate the identification of markers after the recordings and the organization of the saved data. Movements were performed in two joints for each limb, performing the following monoplanar movements:Knee (flexion and extension).Hip (abduction and adduction, flexion and extension).Elbow (flexion and extension).Shoulder (abduction and adduction, flexion and extension).

For each movement, two recordings were made with different inclinations concerning the sensors, 0${}^{\circ }$ and 30${}^{\circ }$, respectively, as shown in Figure 3.

However, the knee flexion movement was performed with the volunteer perpendicular to the sensors so that the activity occurred in the frontal plane of the sensors.

The moving segments were always the right limbs, as shown in Figure 3. In total, 120 repetitions were performed per volunteer and analyzed separately for each sensor.

### 3.4. Data Processing and Treatment

The raw data collected were depth information, spatial marker position, color images, and joint estimates. In this bias, the Kinect V2 provides the joint tracking already included in its SDK, requiring no preprocessing. On the other hand, Astra and RealSense sensors provide joint tracking for a monthly subscription. Because of this, the MediaPipe was used to identify the position of the joints, and then the depth and RGB images were superimposed to obtain the 3D coordinate of the joints.

The joint positions for the four alternatives compared in this work were obtained in the following way:Kinect: The 3D joint positions were directly obtained from Kinect SDK.Astra: We applied MediaPipe’s joint estimation on Astra’s RGB image and then correlated the 2D joint positions with the depth image captured by the sensor, resulting in 3D joint positions.RealSense: We applied MediaPipe’s joint estimation on RealSense’s (D415) RGB image and then correlated the 2D joint positions with the depth image captured by the sensor, resulting in 3D joint positions.MediaPipe: We applied MediaPipe’s joint estimation on Kinect’s RGB image and obtained the estimated 3D joint coordinates (notice that, in this case, the depth information was estimated based on the central point of the user’s body).

#### 3.4.1. Skeleton Processing

After the recording, extracting the skeletons from the recordings is necessary. Each sensor presented a form of skeleton extraction. The Kinect V2 SDK provides the joint estimation. On the other hand, Astra and RealSense do not offer a free charge for the SDK for extraction. Thus, it was necessary for an intermediate extraction step using MediaPipe to identify and track the joints in the 2D RGB images and subsequently perform the overlay with the depth images and convert them into 3D coordinates. On the other hand, for the Qualisys recordings, the company’s licensed software, Visual 3D, was used to determine the positions of the desired joints from the markers.

#### 3.4.2. RGB and Depth Image Overlap

Due to the absence of the free SDK for the Astra Orbbec and Intel RealSense sensors, it was necessary to use a hybrid method for joint inference. The MediaPipe framework (BlazePose) was used to detect and track the volunteers’ joints for the Astra and RealSense devices.

The information in the RGB image is planar coordinates that must be associated with the depth image to obtain the 3D representation of the joints. For this, we used the method described by Knust et. al. [13] of projection and distance similarity relationships.

Knust [13] used the similarity of triangles to correlate the planar measurement with (*x*,*y*) and transform it into 3D world coordinates given the relationships expressed in Equation (Equation 1) and Figure 4.
(1)$$\frac{x_{p}}{d}=\frac{x}{z}$$
(2)$$\frac{y_{p}}{d}=\frac{y}{z}$$

Since the sensor provides depth (*z*) information, the screen coordinates can be transformed into world coordinates. Considering the displacement of the reference origin, we obtain
(3)$$\left(x^{w},y^{w},z^{w}\right)=\left(\frac{(x_{p}-x_{o})*z}{f},\frac{(y_{p}-y_{o})*z}{f},z\right)$$

For the 2D to 3D transformation calculations, a C++ library was implemented, with a Python wrapper, employing PyBind. Thus, C++ was used for Astra since it is the standard language employed in Orbbec’s library. For RealSense, we used its library available in Python, which contained the basic commands for the operation and functioning of the equipment, although it did not include the joint inference SDK.

#### 3.4.3. Angular Calculations

After estimating the positioning of the skeletons, a file was obtained for each motion, and each sensor organized the joints. In general, the files contain the following data organization, ${\left\{{\left[\alpha \right]}_{i}\right\}}_{j}$, where α is a tuple (*x*,*y*,*z*) of the coordinates, i is the index of each joint, and j is the corresponding frame of the recording.

For the angular calculations, the International Society of Biomechanics (ISB) references were adopted for the joint coordinate system (JCS), described by Da Gama [14], using the three anatomical planes (frontal, sagittal, and horizontal) as standard, as shown in Figure 5.

Three joints need to be correlated two by two to calculate the angle of the desired joint being equivalent to the central joint. Then, two successive joints are connected to represent the body segments, with the proximal joint as the origin point and the distal joint as the endpoint, creating a vector.

From the skeletons, the BioAnalysis library was used [13], a library written in Haxe, which uses the ISB standard to calculate the angulation of an input joint or vector relative to one of the body planes [14]. For each frame, the skeleton is passed into the library, along with the joint, plane, and tolerance that will be analyzed (the maximum value that the movement can occur out of the plane). With this, BioAnalysis computes the angle between the normal of the plane and the vector that is moving (if it receives a joint as input, BioAnalysis converts it to the vector that represents the joint).

In practice, first, the skeleton is converted to body coordinates, and the coordinates are given concerning the coordinate system of each joint. To achieve this, each vector undergoes a base change made for its coordinate system. For example, in the case of the upper limbs, the vector from the middle of the column to the top is used as the y-axis, the vector product of the vector created from the left and right hip and the y-axis and the vector product of x and y is usedto form the z-axis.

The coordinate system is normalized and defined by the vectors $\vec{x}$, $\vec{y}$, and $\vec{z}$, described above. The process of converting coordinates from a system centered on the sensor to a system centered on the joint of interest is expressed by the following change-of-basis matrix (Equation (Equation 4)): (4)$${\left[v\right]}_{bc}={\left[I\right]}_{bc}^{sc}{\left[v\right]}_{sc}$$
as long as ${\left[I\right]}_{bc}^{sc}$ is the transformation matrix from sc (sensor coordinates) to bc (body coordinates). Thus, we have
(5)$${\left[I\right]}_{bc}^{sc}={\left\{{\left[B\right]}^{bc}\right\}}^{-1}={\left[\begin{array}{ccc} a_{x} & b_{x} & c_{x}\\ a_{y} & b_{y} & c_{y}\\ a_{z} & b_{z} & c_{z} \end{array}\right]}^{-1}$$

As the vectors in body coordinates, we have $\vec{x}$, $\vec{y}$, and $\vec{z}$ as the vectors (1,0,0), (0,1,0), and (0,0,1), respectively. To calculate the tolerance, first, the angle between the vector that is moving and the normal of the plane in which the movement is being made is analyzed. For this, the mathematical relation of cosine between vectors is used after checking that the vector belongs to the plane, i.e., the vector that is moving must have an angle of 90° to the normal of the plane. Subsequently, considering the given tolerance, the angle between the moving vector and one of the perpendicular vectors that belong to the plane is calculated. For the frontal plane, the calculation between the moving vector and the y-axis will be performed. An error is returned if the vector does not belong to the plane.

#### 3.4.4. Data Treatment

After computing the angles, the motion signals must be smoothed and synchronized. A Python script was built using the libraries NumPy, SciPy, and Matplotlib because they are free and allow easy adjustments and the possibility to share the algorithm.

In the data processing flow, first, filtering is performed with the seventh-order Butterworth low-pass filter with a cutoff frequency of 5 Hz and 30 Hz for the signals from the RGB-D and Qualisys sensors. Then, a moving average filter with a kernel size of 7 is applied.

With the signals properly filtered and better smoothed, it is necessary to make them of equal size since the signals coming from Qualisys present more than 6 times the capture frequency of the RGB-D sensors (180 Hz/30 Hz). Therefore, the interpolation process was carried out simultaneously with the synchronization process. The signals were divided into three parts:Start to first peak.First peak to last peak.Last peak.

The peaks of the signals represent the same instant of time, which is the maximum angular displacement value of the movement under analysis. First, the distance between the number of frames between the samples about the interpolation reference was identified. Then, we used a second-order polynomial interpolation function from these sections and the number of points that needed to be interpolated.

The first process was to perform peak identification to identify the beginning of the movements. Figure 6 displays the change in each piece of data throughout the process.

Figure 7 displays the difference in sample sizes resulting from the discrepancy between the sample frequencies of the equipment. After interpolation and synchronization, all data presented the same phase (Figure 8). It is important to note that, even with the filters, the curves of the RGB-D equipment did not show the same smoothness as those of the Qualisys.

An alphabetic code was determined to simplify the recording and identification of movements, following the rule: Joint-movement-position relative to the sensor index of the recording, K-FLEX-FR-1 stands for knee-flexion-frontal-recording 1.

#### 3.4.5. Validation and Statistical Variables

With the data adequately interpolated and synchronized, it was possible to perform the comparison tests. As the objective of this work was to evaluate the performance of different sensors in relation to the gold standard, the sensors’ absolute error (AE) and their root mean square (RMS) were calculated to understand the error variation and sample dispersion.
(6)$$E_{A}=\left|\theta_{s}-\theta_{Q}\right|$$
(7)$${RMS_{A}}_{E}=\sqrt{\frac{1}{n}\left(x_{1}^{2}+x_{2}^{2}+\cdots +x_{n}^{2}\right)}$$

The absolute error was classified as excellent, good, moderate, or poor according to its median value and standard deviation concerning the clinical reference (CR), estimated by the human error performed in the clinical environment (12.78 ± 7.44 degrees) [15]. Thus, the final classification was as follows: (i) great, when error plus its standard deviation is below CR; (ii) good, if the error is below the CR; (iii) moderate, if it crosses CR until five degrees; and (iv) weak, if the error is more than 5 degrees higher than CR.

In addition, we evidenced that the data did not behave as a normal distribution, thus we can classify them as a nonparametric dependent. Since the data distribution did not behave parametrically, the median data translates more behavioral information than the mean. Along with the median, the RMS represents the study’s dispersion of the absolute error data. Therefore, we used the significance of the difference test, Friedman’s rank-sum analysis, for multiple comparisons of the sensors, with the same levels for *p* < 0.05 and (**) for *p* < 0.001. Another post hoc analysis used was the Wilcoxon test between each pair of sensors, where two levels of significance were defined and reported as (*) for *p* < 0.05 and (**) for *p* < 0.001 [16].

To finalize the analysis, Pearson’s correlation was used between the angular values because the correlation indicates the relative proximity between two variables and the strength of their linear relationship. We used the Pearson correlation scale established by Portney and Watkins. The correlation limits are classified as poor (<0.5), moderate (≥0.5 and <0.75), good (≥0.75 and <0.9), and excellent (≥0.9) [17].

The absolute error, RMS, and Pearson’s correlation between the gold standard and the sensors were evaluated for the entire dataset, and each movement was analyzed. R software was used for the statistical calculations, as it is free and open source.

## 4. Results

This section contains the results obtained from the processes described for data treatment. A quantitative and statistical evaluation of the angular information was performed to determine the equipment’s accuracy compared to the gold standard. For better visualization and understanding of the results, this section was divided into data loss, general results, lower limbs, and upper limbs.

### 4.1. Data Loss

Some of the movements listed for the recordings could not be compared due to the infeasibility of extracting joints in Visual 3D and/or consistency in the RGB-D sensor recordings. This problem was due to the loss of joint definition markers throughout the recording, so it was impossible to interpolate these markers’ displacement. Thus, it was impossible to define the segments in Visual 3D to determine the position of the joints. Additionally, sensor losses occurred due to generating noisy data or asynchronous depth and color image recordings. When the loss came from one sensor, the other recordings were not evaluated to ensure comparison among all sensors at all times. As shown in Table 3, we could not analyze 19 from 120 recordings; 15.83% lost. In addition, the 101 was scrutinized, resulting in 505 execution.

### 4.2. General Data

Of the variables analyzed, the first was the absolute error of the angular variation of the sensors against Qualisys. In Table 4, it is possible to see the error for all movements and split for upper and lower limbs movements. In addition, it is possible to analyze its classification in relation to human error performed in clinical practice (12.78 ± 7.44 degrees) [15].

The absolute error was classified as excellent, good, moderate, and poor according to its median value and standard deviation concerning the clinical reference (CR), estimated by the human error performed in the clinical environment (12.78 degrees) [15]. Thus, the final classification was (i) excellent when error plus its standard deviation is below CR; (ii) good if the error is below the CR; (iii) moderate if it crosses CR until five degrees; and (iv) poor if the error is more than 5 degrees higher than CR.

It is possible to see the error distribution. Among the sensors, MediaPipe presented the lowest median (8.3${}^{\circ }$). Kinect showed the highest median value, 12.2${}^{\circ }$. Astra and Intel gave higher maximum and peak values of discrepancy.

Evaluating Figure 9, it is possible to understand that the behavior of MediaPipe and Kinect presented less dispersed values. This dispersion can be considered by the RMS values of the absolute error. Given this, it is possible to notice that the distribution of error values for the Kinect and MediaPipe sensors obtained the lowest values.

The last element evaluated was the correlation between the sensors and Qualisys by Pearson’s method. In the Pearson correlation matrix test, all sensors presented a p<0.001. Table 5 displays the Pearson correlation values for the sensors; MediaPipe and Kinect had good correlation values, while Astra and Intel had moderate correlation values.

### 4.3. Lower Limbs

The separate evaluation of the lower limbs was performed in two forms of analysis, first considering all the LE data and later each movement and evaluating the mean.

#### 4.3.1. General Analysis

Figure 10 expresses the box plot for the absolute error of the samples in a paired side-by-side comparison of the sensors. The results were very close to the overall comparisons in the previous subtopic. MediaPipe showed the lowest median absolute error, a value of 7.01°. At the same time, Kinect had the second lowest median, 8.56${}^{\circ }$. On the other hand, Astra and Intel performed with the highest median values for error, respectively, 9.86${}^{\circ }$ and 9.9${}^{\circ }$. These values indicate that the error rate within the distribution presents the middle of the matters at these points. Associating this with the RMS, it is possible to understand, along with the box plot, the dispersion behavior of the error within the sample. Figure 11 shows the RMS values, and it is possible to see that MediaPipe and Kinect presented less dispersion. However, it is common for all sensors to have high maximum error values, possibly resulting from synchronization or unfiltered noise.

After determining statistical relationships and error results, data correlation was calculated with Qualisys to determine the sensor closest to the gold standard. Table 6 shows Pearson’s correlation between RGB-D equipment in ascending order from left to right, proximity range from 0 to 1. Kinect and MediaPipe both obtained a good index.

#### 4.3.2. Motion Analysis

The same analysis as in the previous section was performed for each type of movement recorded to verify the performance at each joint and recording slope. Through the statistical tests, it was determined that the samples do not behave as normalized data. Therefore, the post hoc nonparametric Friedman rank sum was used, obtaining the value of *p* < 0.001 for all sensors.

The absolute error and the RMS were calculated using the same analysis methods for each movement. Figure 12 condenses the information of the AE distribution. Evaluating the results for each movement, we can see that in Q-ADB-FR the Astra sensor had the lowest median value (5.39${}^{\circ }$); in Q-FLEX-FR, Kinect had the lowest median value for the error (4.28${}^{\circ }$); for the other movements, MediaPipe presented the lowest error values, being J-FLEX-FR (6.74${}^{\circ }$), J-FLEX-INL (8.56${}^{\circ }$), abd Q-ADB-FL. In addition, the Q-ADB-FR and Q-FLEX-FR movements had the second smallest values, respectively, 5.63${}^{\circ }$ and 7.67${}^{\circ }$.

Table 7 presents the values described for each movement to understand all the movements and median error values. MediaPipe had the lowest mean median as well as the standard deviation. Kinect V2 sensor had the second lowest mean and standard deviation of the median absolute error. Since the data are not parametric, the evaluation of the median indicates the data behavior; associating it with the RMS makes it possible to establish and understand the data behavior and evaluate which one presented the most desired behavior.

Evaluating the RMS of the $E_{A}$ of the movements together with the box plots, it is possible to understand the behavior of the data in comparison with the gold standard and to have indications of which presented a better performance in relation to the error. Therefore, Figure 13 contains a side-by-side comparison of all the movements in a bar chart. There is a match between the RMS and the box plot. Five of the six smallest RMS values coincided with being the same sensor with the smallest medians. Although they are complementary information, they are not correlated. A discrepancy occurred only in the Q-ABD-FR movement since the lowest median value was from Astra and the RMS from MediaPipe. This occurred due to the dispersion of the data of the first and third quartile and higher maximum, compared to the box plot of MediaPipe.

Table 8 presents the average behavior of the RMS along the movements. Again, MediaPipe obtained the lowest average value and the standard deviation. Thus, it can be interpreted that the MediaPipe data presented less dispersion among the sensors for lower limbs.

The previous subsection performed the Pearson correlation test for each movement. Table 9 displays the Pearson correlation between the RGB-D equipment for all sensors and movements, the value of *p* < 0.001 in the correlation matrices. Kinect V2 obtained a correlation index of excellent in two movements (Q-ABD-INL and Q-FLEX-FR), being the only one to achieve this metric. However, in the average evaluation, Kinect and MediaPipe obtained an index of good.

### 4.4. Upper Limbs

The absolute error, RMS, and Pearson’s correlation between the gold standard and the sensors were evaluated for the entire dataset and for each movement analyzed. The process was the same for the upper limbs, making it possible to present the results more concisely.

#### Motion Analysis

As in the previous section, the data were analyzed by Friedman’s rank sum test *p*-value < 0.001 considering all sensors for the comparison. The absolute error and RMS were calculated using the same analysis methods for each motion. Figure 14 condenses the information of the $E_{A}$ distribution. In C-FLEX-FR, Kinect obtained the lowest median with 10.26${}^{\circ }$. In C-FLEX-INL, Intel obtained the smallest with 9.67${}^{\circ }$. Astra obtained the lowest median values in two moves: O-ABD-FR, 7.5${}^{\circ }$, and O-ABD-INL, 7.48${}^{\circ }$. Similarly, MediaPipe showed lower values in two movements: O-FLEX-FR, 10.26${}^{\circ }$, and O-FLEX-INL, 6.81${}^{\circ }$.

Table 10 presents the values described for each movement to understand all the movements and median error values. MediaPipe presented the lowest average median, followed by Intel, which presented a small standard deviation compared to the others. The lowest median indicates the behavior, representing that in half of the data, the error was at least lower than the median, its lowest value being a positive indicator.

In addition, to evaluate the behavior of the error dispersion, it is possible to associate it with the RMS. Figure 15 presents the RMS results of the absolute error. Kinect presented lower RMS values in C-FLEX-FR and C-FLEX-INL. The second one presented the second lowest median (11.13${}^{\circ }$) but with smaller data dispersion. In O-ABD-INL, Intel presented the lowest RMS with 13.17${}^{\circ }$, followed directly by MediaPipe with 13.86${}^{\circ }$. MediaPipe obtained the lowest values in both movements where it obtained a lower median, O-FLEX-FR, 12.19${}^{\circ }$, and O-FLEX-INL, 17.33${}^{\circ }$, in addition to O-ABD-FR, with RMS of 10.94${}^{\circ }$.

Additionally, Table 11 expresses the RMS across the movements and the average by comparing the results. MediaPipe obtained the lowest average RMS and standard deviation, followed by the Kinect. Therefore, the less dispersed behavior of the MediaPipe is noticeable.

As in the previous section, Pearson’s correlation test was performed for each movement. Table 12 displays Pearson’s correlation between the RGB-D equipment. For upper limbs, the correlation between RGB-D and RGB equipment was high, as the lowest correlation obtained the metric of good. Kinect and MediaPipe achieved excellent correlation.

## 5. Discussion

The objective of the work was to evaluate the analysis capacity of RGB-D equipment as a source of joint position estimation for biomechanical evaluation. Consequently, assessing the equipment’s ability to recognize and track the joints’ 3D position enables the measurement of angular variations as long as the range of motion is a parameter used to follow the evolution and condition of the patient [18]. The Kinect V2 sensor and the MediaPipe API showed better results in the metrics evaluated. On the other hand, Astra and Intel showed inferior performances.

Along with the median, the RMS represents the study’s dispersion of the absolute error data. Thus, MediaPipe obtained the best absolute error and RMS results so that the median error was lowest on average in the analyses of MMSS ($E_{A}=9.98\pm 3.79$ and RMS=16.45±4.93) and MMII ($E_{A}=7.16\pm 1.21$ and RMS=16.92±3.39). Considering all recordings for the limbs, MediaPipe performed wellwith the results of $E_{A}=8.57\pm 3.06$ and RMS=16.68±4.05.

In this bias, it is necessary to evaluate the error in front of other forms of measurement. In their work, Hancock et al. [19] evaluated different conditions and equipment of goniometry common in clinical settings for measuring knee angle in flexion–extension. In their work, Halo digital goniometer, long- and short-arm goniometer, visual estimation, and cell phone app goniometer were evaluated, obtaining the estimated error for each piece of equipment, respectively: 6${}^{\circ }$; 10${}^{\circ }$; 14${}^{\circ }$, 14${}^{\circ }$, and 12${}^{\circ }$. Therefore, it can be seen that the MediaPipe error would be below the measurement range of short- and long-arm goniometers (error between 10${}^{\circ }$ and 14${}^{\circ }$), while the other sensors would be within the range. Furthermore, MediaPipe’s average error of 8.57${}^{\circ }$ is below the error of tools commonly used in physical therapy clinics for joint angle assessment. Additionally, compared to the results obtained by Russo [15], the human error for angular measurements is 12.78 degrees. Thus, MediaPipe scored excellent for all data and lower limbs and good metric for upper limbs. Kinect scored excellent for lower limbs, good for all data, but a moderate score for upper limbs, the lowest result among the sensors.

It is a tool that uses only one RGB camera and can estimate depth data and thereby calculate three-dimensional angular variations. Thus, the performance of the MediaPipe was surprisingly better than three RGB-D devices (Astra, Intel, and Kinect V2) for measuring angular variation.

The Astra and Intel sensors had the worst correlation and numerical results for lower limbs. Both showed moderate results of r and the highest results of absolute error and RMS, values far away from the other sensors. Kinect v2 and MediaPipe showed good correlation results. MediaPipe showed results closer to the standard ($E_{A}=12.51\pm 11.33$ and RMS=17.22), while Kinect showed intermediate results between the sensors of $E_{A}=14.21\pm 15.81$ and RMS=21.26.

The upper limbs results are similar concerning the LE, since the Kinect and MediaPipe perform better than Astra and Intel RealSense. As seen in Table 6, Table 9 and Table 12, this correlation was better for all data, with the sensors obtaining at least moderate correlation. Astra and RealSense showed a correlation of good. For AE and RMS values, both showed worse results, although, better than in LE, Astra obtained $E_{A}=16.9\pm 16.50$ and RMS=23.65, while Intel obtained $E_{A}=15.58\pm 14.81$ and RMS=21.50. We did not find in the literature articles evaluating these devices’ performances of angular metrics.

Furthermore, the Kinect and the MediaPipe showed better ability to evaluate angular measurements. When each type of movement was assessed, both showed excellent correlation. Mangal and Tiwari [20], in their work on muscle dysfunction assessment, found excellent correlation values for shoulder abduction and flexion movements, respectively, r = 0.98 and r = 0.96, corroborating the research data.

This work presented some limitations in its application. First, the small number of volunteers did not address the diversity of body pattern variations to evaluate the capabilities of different anatomical characteristics. The low number was due to movement restrictions and social distance because of COVID-19. Third, the high number of repetitions was established to obtain a considered number of executions of the analysis movement. In this movement bias, the synchronization was performed by identifying the movement peaks and not by synchronizing the start of several equipment. This option facilitated the collection asthe sampling theorem states that for the reconstruction of a system, it is necessary to capture at least twice the frequency; thus, recording at 30 Hz guarantees the rebuilding of the data entirely once the movements had no more than 1 Hz execution. Finally, front camera testing in monoplanar movements may have made it easier to estimate the depth measurement of the MediaPipe, whereas the inclined recordings were performed.

## 6. Conclusions

The application of motion capture and analysis techniques represents powerful tools in the rehabilitation field and can bring several advantages to patient treatment. However, applying optical MoCaps of reflective markers widely in clinical settings is limited due to their high cost and the high demand for time and skilled labor for setup. In addition, considering the context experienced in the pandemic, there was, in many cases, the impossibility or the need to remove the patient from treatment centers due to the risk of contamination. Thus, the need for remote monitoring was evident in a reliable way for the continuity of motor rehabilitation treatments.

In this perspective of use in clinical and even home care environments, RGB-D and RGB applications may represent a paradigm shift in healthcare. First, however, it is necessary to understand how close these devices are to the gold standard. Because of this, this study sought to analyze the ability to evaluate and estimate joint angles. We investigated three RGB-D equipment (Kinect V2, Astra, and Intel RealSense) and a technique based on RGB images (MediaPipe). Among the results, the Intel and Astra equipment had the worst correlation values with the gold standard; however, they were within the error range of goniometry equipment in all evaluations. On the other hand, Kinect had one of the best correlation results and the second lowest error for lower limbs. Still, its performance in the upper limbs was the worst, making its overall average error the highest.

In contrast, MediaPipe showed the best values. The Kinect V2 result corroborates the literature data with correlations of good and excellent for angular evaluation. In addition, the MediaPipe showed a surprising result, presenting numerical results better than the other sensors and below the error scales of clinical equipment. Thus, this represents the possibility of developing and applying techniques using only RGB components for rehabilitation applications on par with RGB and depth imagery solutions.

The results of the research demonstrate the feasibility of applying the MediaPipe for the measurement of angular variation for biomechanical evaluation. Therefore, the following steps of the study will be to actively search for other RGB methods that present depth calculation and to perform tests in other, more complex, and functional activities and examinations with users in a clinical environment.

## Figures and Tables

**Figure 1 sensors-23-00003-f001:**
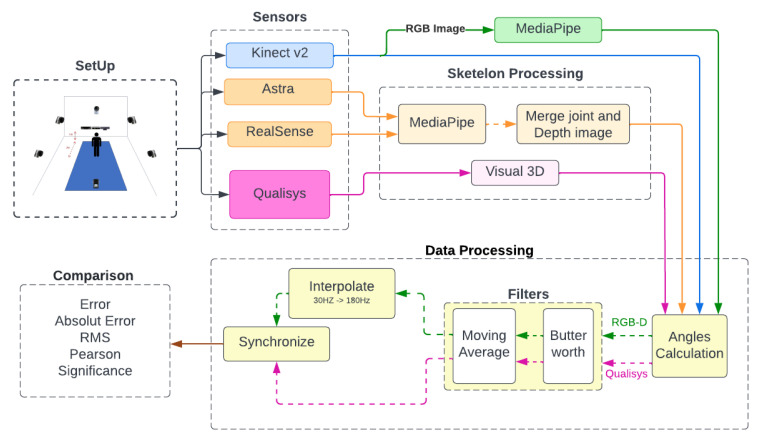
General flowchart of the steps and processes performed in the research.

**Figure 2 sensors-23-00003-f002:**
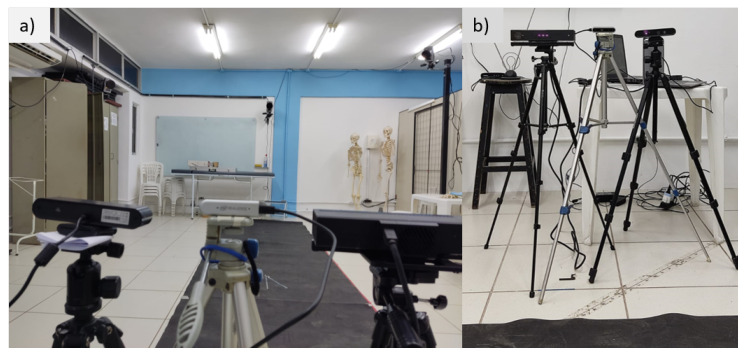
View of the RGB-D equipment. (**a**) Posterior view of the equipment aligned with the recording track. (**b**) Front view of the equipment and lifting tripods.

**Figure 3 sensors-23-00003-f003:**
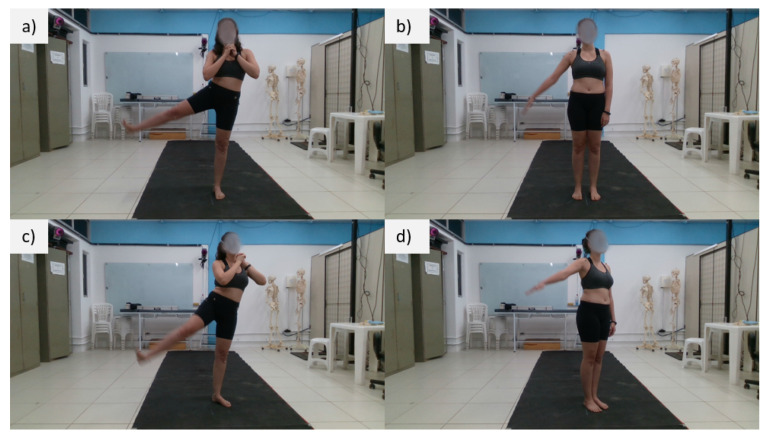
Movements and positioning of the volunteers during the exercises. (**a**) Movement of the LE facing the sensors. (**b**) Movement of the UE facing the sensors. (**c**) Movement of the LE inclined to the sensors. (**d**) Movement of the UE inclined to the sensors.

**Figure 4 sensors-23-00003-f004:**
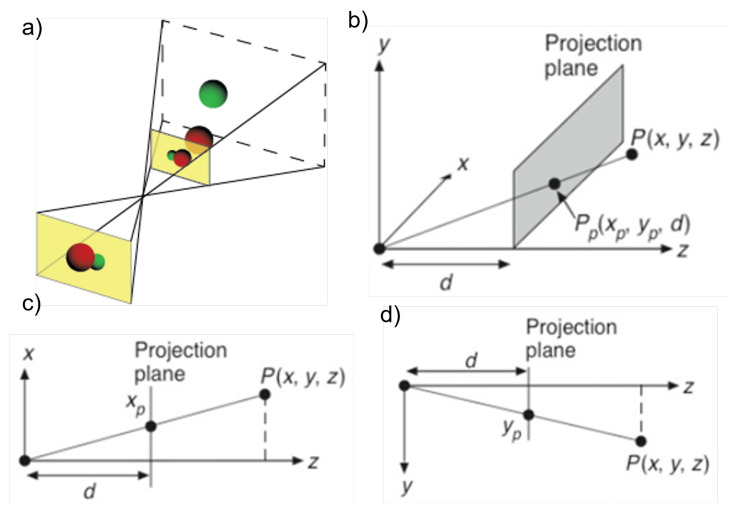
Representation of the angular similarity process. (**a**) Projection of the front of the camera. (**b**) Representation without considering main points. (**c**) The x-axis view. (**d**) The y-axis view [13].

**Figure 5 sensors-23-00003-f005:**
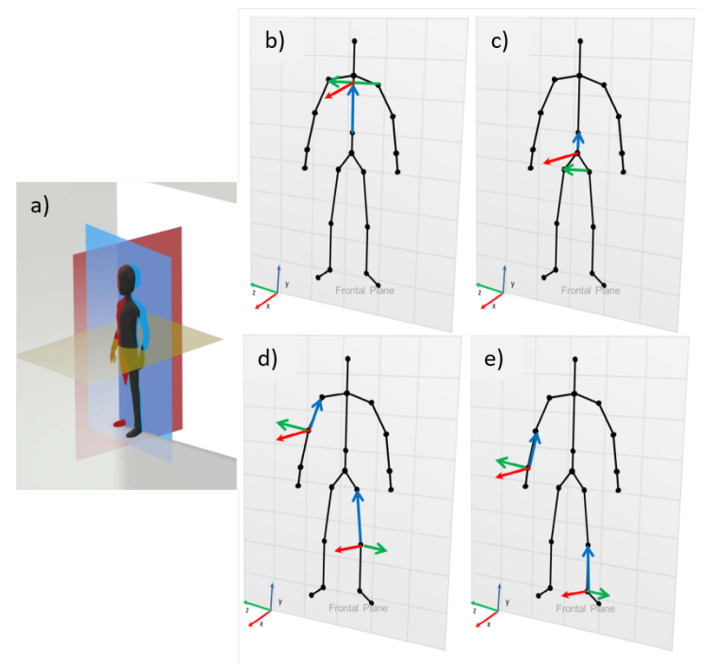
Representations of ISB definitions. (**a**) Anatomical planes. References used for the coordinate system: (**b**) shoulder and cervical; (**c**) pelvis and lumbar; (**d**) elbow and knee; (**e**) wrist and ankle. Adapted from [14].

**Figure 6 sensors-23-00003-f006:**
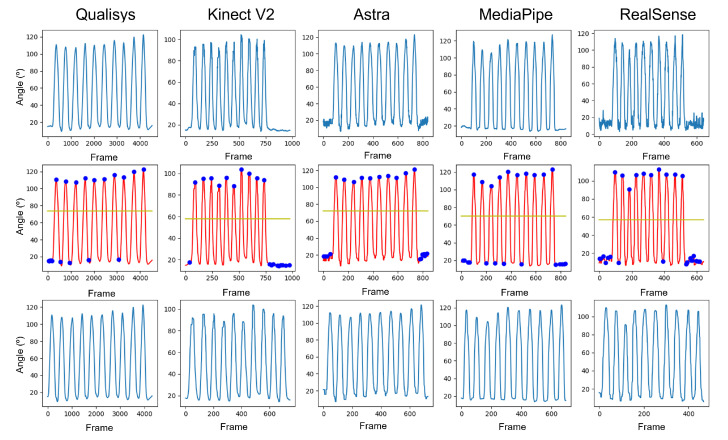
Graphs of the data treatment process. First row contains the pure data without any kind of treatment. Second row represents the data after filtering with the moving average and Butterworth filters. Third row contains data with only the motion execution.

**Figure 7 sensors-23-00003-f007:**
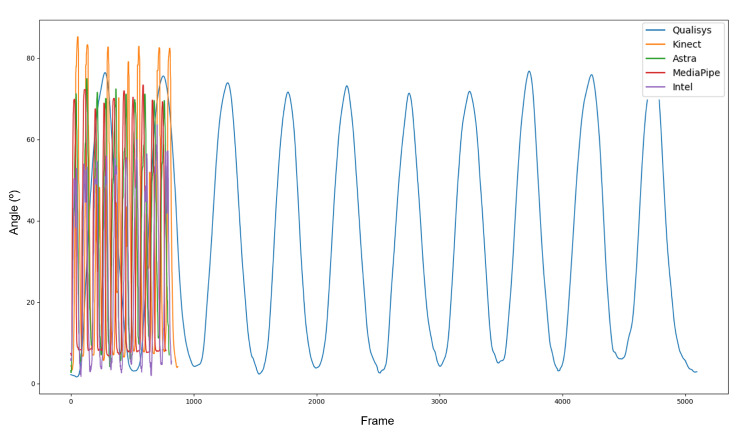
Unsynchronized and interpolated data.

**Figure 8 sensors-23-00003-f008:**
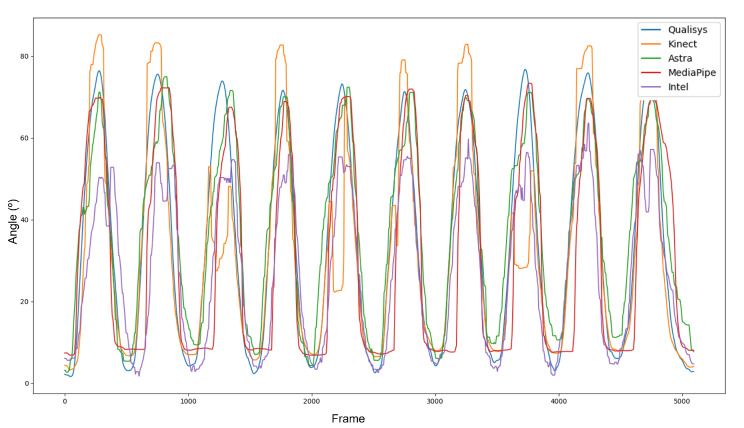
Synchronized and interpolated data.

**Figure 9 sensors-23-00003-f009:**
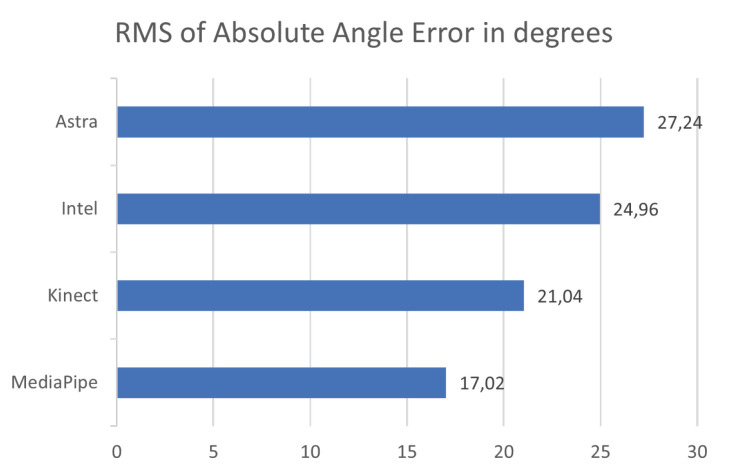
Absolute error RMS plot comparing all sensors for error dispersion.

**Figure 10 sensors-23-00003-f010:**
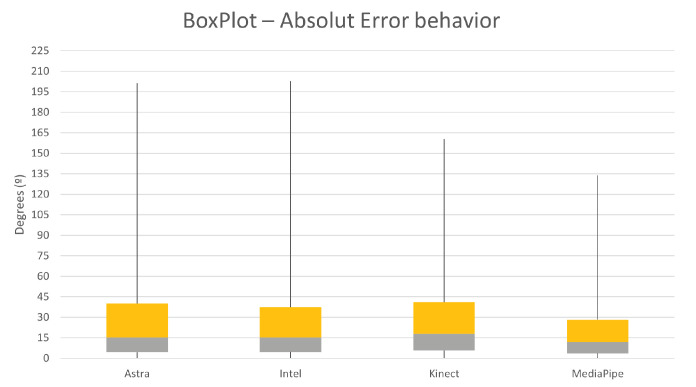
Box plot of the absolute error of each sensor in relation to Qualisys, for the lower limb movement.

**Figure 11 sensors-23-00003-f011:**
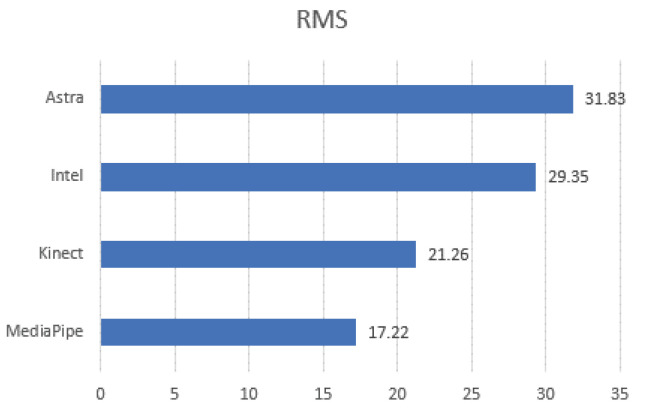
RMS of the absolute error for each sensor against all lower limb data in degrees.

**Figure 12 sensors-23-00003-f012:**
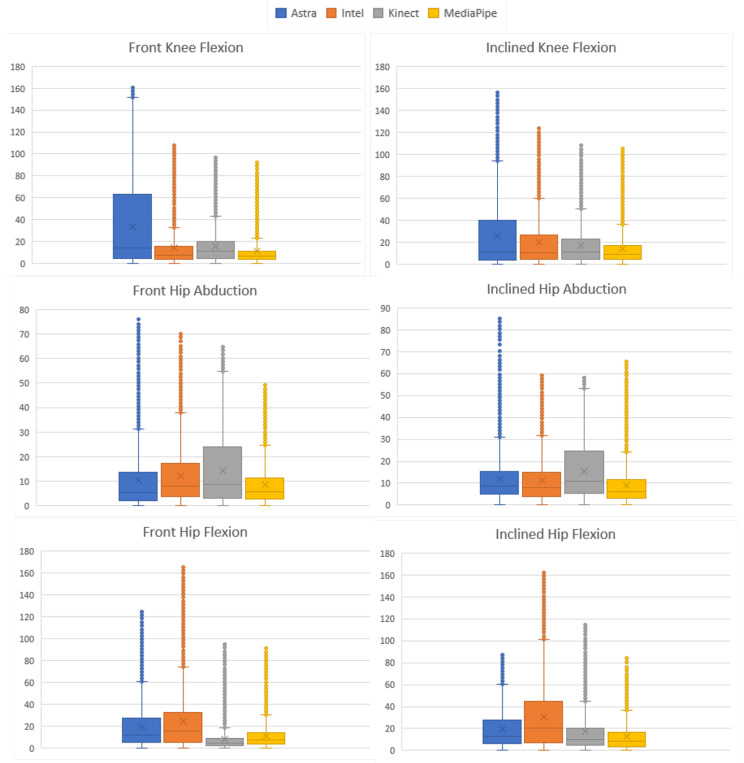
Grouped box plots of the absolute error (in degrees) of each LE movement.

**Figure 13 sensors-23-00003-f013:**
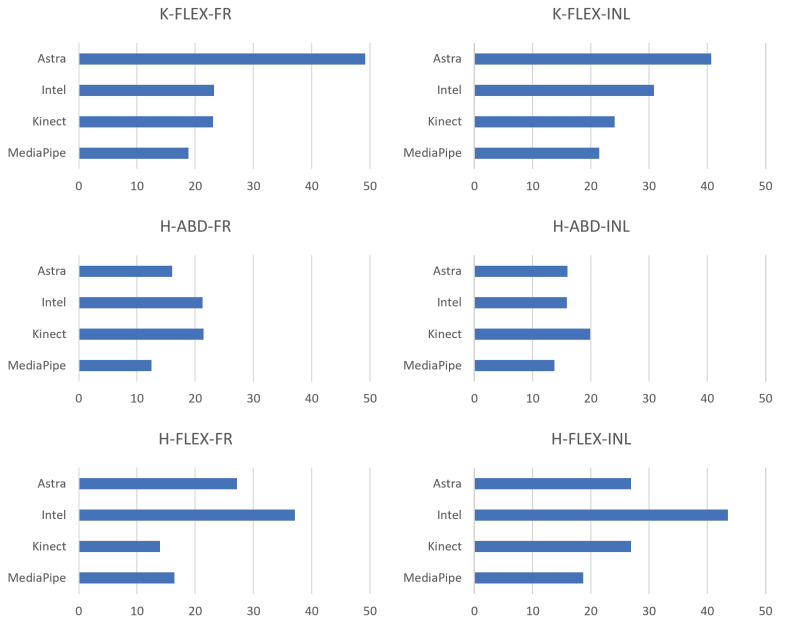
RMS plots of the pooled absolute error (in degrees) for all sensors for LE collection.

**Figure 14 sensors-23-00003-f014:**
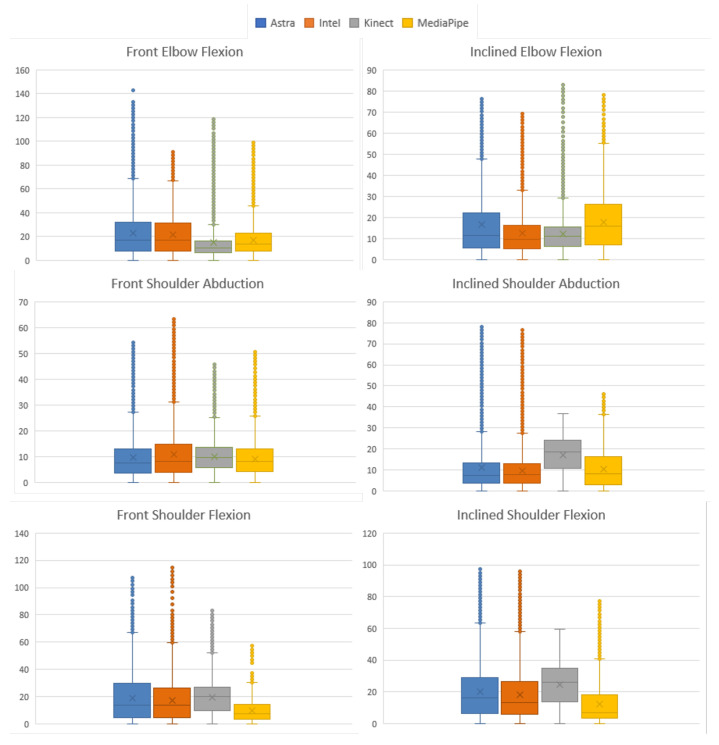
Grouped box plots of the absolute error (in degrees) of each UE movement.

**Figure 15 sensors-23-00003-f015:**
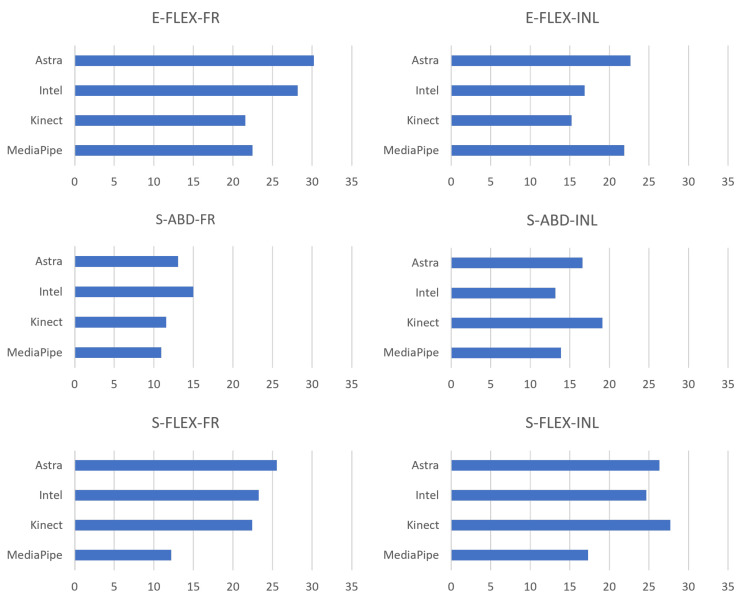
RMS plots of the pooled absolute error (in degrees) for all sensors for UE collection.

**Table 1 sensors-23-00003-t001:** Volunteer characteristics.

	Mean	Standard Deviation	Minimum	Maximum
Age [y]	21.5	1.37	19	23
Height [m]	1.69	0.12	1.65	1.96
Weight [kg]	63.65	7.33	55	73
BMI [kg/cm${}^{2}$]	21.17	2.8	18.22	25.26

**Table 2 sensors-23-00003-t002:** Specification of the equipment used.

vProperties	Kinect v2	Astra	RealSense
RGB camera resolution	1920 × 1080 (px)	640 × 480 (px)	1920 × 1080 (px)
Depth camera resolution	512 × 424 (px)	640 × 480 (px)	1280 × 720 (px)
Frame rate	30 (fps)	30 (fps)	30 (fps)
RGB field of view	70 × 60 (degrees)	60 × 49.5 (degrees)	69 × 42 (degrees)
Depth field of view	84 × 53 (degrees)	60 × 49.5 (degrees)	69 × 42 (degrees)

**Table 3 sensors-23-00003-t003:** Table of recorded movements, explaining which ones were not analyzed, identified by “o” for the positive ones and “x” for the negative ones.

	Volunteer 1	Volunteer 2	Volunteer 3	Volunteer 4	Volunteer 5
K-FLEX-FR-1	o	x	o	o	o
K-FLEX-FR-2	o	x	o	o	o
K-FLEX-IN-1	o	o	o	o	x
K-FLEX-IN-2	o	o	o	o	o
H-ABD-FR-1	o	o	x	o	o
H-ABD-FR-2	o	o	o	o	o
H-ABD-IN-1	o	x	o	o	x
H-ABD-IN-2	o	x	o	o	o
H-FLE-FR-1	o	o	o	o	o
H-FLE-FR-2	o	o	x	o	o
H-FLEX-IN-1	x	o	o	o	o
H-FLEX-IN-2	x	o	o	x	o
E-FLEX-FR-1	o	o	o	o	x
E-FLEX-FR-2	o	o	o	o	o
E-FLEX-IN-1	o	o	o	x	o
E-FLEX-IN-2	x	o	o	o	x
S-ABD-FR-1	x	o	o	x	o
S-ABD-FR-2	o	x	o	o	o
S-ABD-IN-1	o	o	o	o	o
S-ABD-IN-2	x	o	o	o	o
S-FLEX-FR-1	o	o	o	o	o
S-FLEX-FR-2	o	o	o	o	o
S-FLEX-IN-1	o	o	o	o	o
S-FLEX-IN-2	o	o	o	o	o

**Table 4 sensors-23-00003-t004:** Absolute error classification in relation to clinical error. Excellent is in dark green, good is in light green, moderate is in yellow, and poor is in red. Data are presented as means and standard deviation in degrees (${}^{\circ }$) for all movements and splitting for upper limbs and lower limbs movements.

Sensor/Data	All Data	Upper Limbs	Lower Limbs
Astra	11.60 ± 3.71	12.36 ± 4.27	10.84 ± 3.27
Intel	11.56 ± 4.38	11.56 ± 3.74	11.57 ± 5.32
Kinect	12.65 ± 5.99	16.01 ± 6.72	9.30 ± 2.62
MediaPipe	8.57 ± 3.06	9.98 ± 3.79	7.16 ± 1.21

**Table 5 sensors-23-00003-t005:** Pearson correlation for the angular variation data. For all sensors, *p* < 0.001.

Astra	Intel	Kinect	MediaPipe
0.69	0.73	0.81	0.86

**Table 6 sensors-23-00003-t006:** Pearson correlation of sensors for all data to LL data; blue (>0.75 and <0.9) and yellow (>0.5 and <0.75).

Astra	Intel	Kinect	MediaPipe
0.62	0.66	0.80	0.87

**Table 7 sensors-23-00003-t007:** Table of medians of absolute error for lower limbs in degrees (${}^{\circ }$).

	Astra	Intel	Kinect	MediaPipe
K-FLEX-FR	14.34	7.3	10.74	6.74
K-FLEX-INL	11.3	10.26	11.38	8.56
H-ABD-FR	5.39	8.08	8.86	5.63
H-ABD-INL	8.79	7.79	10.89	6.03
H-FLEX-FR	11.97	15.46	4.29	7.67
H-FLEX-INL	13.22	20.5	9.64	8.31
Mean	10.84±3.26	11.57±5.32	9.3±2.62	7.16±1.21

**Table 8 sensors-23-00003-t008:** Absolute error RMS values and mean for lower limbs in degrees (${}^{\circ }$).

	Astra	Intel	Kinect	MediaPipe
K-FLEX-FR	14.34	7.3	10.74	6.74
K-FLEX-INL	11.3	10.26	11.38	8.56
H-ABD-FR	5.39	8.08	8.86	5.63
H-ABD-INL	8.79	7.79	10.89	6.03
H-FLEX-FR	11.97	15.46	4.29	7.67
H-FLEX-INL	13.22	20.5	9.64	8.31
Mean	29.31±13.32	28.63±10.44	21.53±4.42	16.9 ± 3.39

**Table 9 sensors-23-00003-t009:** Pearson correlation of the sensors for each lower limb movement; *p*-value < 0.001 for all Pearson correlationmatrices.

	Astra	Intel	Kinect	MediaPipe
K-FLEX-FR	0.31	0.83	0.83	0.89
K-FLEX-INL	0.51	0.69	0.81	0.86
H-ABD-FR	0.54	0.26	0.82	0.69
H-ABD-INL	0,7	0.37	0.91	0.66
H-FLEX-FR	0.68	0.49	0.91	0.87
H-FLEX-INL	0.71	0.41	0.69	0.84
Mean	0.58 ± 0.15	0.51 ± 0.22	0.83 ± 0.08	0.8 ± 0.1

**Table 10 sensors-23-00003-t010:** Grouped box plots of the absolute error of each UE movement.

	Astra	Intel	Kinect	MediaPipe
E-FLEX-FR	17.31	17.23	10.26	13.56
E-FLEX-INL	11.6	9.7	11.13	15.92
S-ABD-FR	7.5	8.3	9.6	8.17
S-ABD-INL	7.8	7.9	18.59	7.97
S-FLEX-FR	13.88	13.58	20.31	7.47
S-FLEX-INL	16.39	13.18	26.15	6.81
Mean	12.36±4.27	11.56±3.74	16.01±6.74	9.98±3.79

**Table 11 sensors-23-00003-t011:** RMS values of absolute error and mean for upper limbs.

	Astra	Intel	Kinect	MediaPipe
E-FLEX-FR	30.24	28.20	21.55	22.49
E-FLEX-INL	22.65	16.86	15.21	21.87
S-ABD-FR	13.05	14.98	11.56	10.94
S-ABD-INL	16.61	13.17	19.11	13.87
S-FLEX-FR	25.56	23.26	22.43	12.19
S-FLEX-INL	26.34	24.66	27.68	17.33
Mean	22.41±6.45	20.19±6.01	19.59 ± 5.68	16.45±4.93

**Table 12 sensors-23-00003-t012:** Pearson correlation of the sensors for each upper limbs movement.

	Astra	Intel	Kinect	MediaPipe
E-FLEX-FR	0.60	0.63	0.83	0.78
E-FLEX-INL	0.73	0.87	0.93	0.85
S-ABD-FR	0.91	0.87	0.96	0.97
S-ABD-INL	0.84	0.90	0.96	0.96
S-FLEX-FR	0.73	0.81	0.91	0.96
S-FLEX-INL	0.82	0.76	0.94	0.95
Mean	0.77 ± 0.11	0.81 ± 0.10	0.92 ± 0.05	0.91 ± 0.08

## Data Availability

Not applicable.

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
