# Peer review of "Validation of Angle Estimation Based on Body Tracking Data from RGB-D and RGB Cameras for Biomechanical Assessment"

_sensors, 2022, doi:10.3390/s23010003_

Round 1
Reviewer 1 Report
As shown in the following web page, there are a number of Intel RealSense depth cameras that are commercially available. The authors should provide the model number for their Intel RealSense sensor.
I believe the main contribution of this work is to demonstrate the efficacy of MediaPipe by comparing it to the gold standard and devices such as Kinect. The results are useful and valuable. However, the authors should clarify how their MediaPipe results were obtained. As shown in Figure 1, MediaPipe was used for Kinect, RealSense, and Astra. Did the results of MediaPipe presented in this work were obtained by processing the RGB images of Kinect? Did MeidaPipe do the same thing for Astra and RealSense? The authors should clearly describe how their best results (which were obtained by MediaPipe) were obtained.
Once joints have been identified, one can readily determine joint angles. If this is the case, please clarify the necessity of obtaining body skeletons.
Considering the power of MediaPipe, it seems that the task of joint tracking can also be effectively accomplished by using a smartphone to acquire RGB images. The authors should comment on this possibility to help the readers to better understand the potential and limitations of MediaPipe.
Author Response
Dear Reviewer,
We really appreciate all the suggestions and comments regarding our paper. We would like to say that we carefully considered all suggestions and items pointed out. We describe how we dealt with each of the points highlighted by the two reviewers as follows.
Reviewer 1:
Reviewer comment: As shown in the following web page, there are a number of Intel RealSense depth cameras that are commercially available. The authors should provide the model number for their Intel RealSense sensor.
Author's comment: The specification of the model used (D415) was added to the text.
Reviewer comment: I believe the main contribution of this work is to demonstrate the efficacy of MediaPipe by comparing it to the gold standard and devices such as Kinect. The results are useful and valuable. However, the authors should clarify how their MediaPipe results were obtained. As shown in Figure 1, MediaPipe was used for Kinect, RealSense, and Astra. Did the results of MediaPipe presented in this work were obtained by processing the RGB images of Kinect? Did MeidaPipe do the same thing for Astra and RealSense? The authors should clearly describe how their best results (which were obtained by MediaPipe) were obtained.
Author's comment: We have added a paragraph in the "Data Processing and Treatment" section detailing how the 3D joint positions were obtained for each of the options compared (Kinect, Astra, RealSense and MediaPipe).
Reviewer comment: Once joints have been identified, one can readily determine joint angles. If this is the case, please clarify the necessity of obtaining body skeletons.
Author's comment: We have added this description in Material and methods section to clarify what "skeleton extraction/processing" means in the text: "By skeleton extraction, we mean extracting the 3D positions of the human body joints.". Indeed, once the joint positions are estimated, we can calculate the angles between them according to the definition of each joint.
Reviewer comment: Considering the power of MediaPipe, it seems that the task of joint tracking can also be effectively accomplished by using a smartphone to acquire RGB images. The authors should comment on this possibility to help the readers to better understand the potential and limitations of MediaPipe.
Author's comment: We have added this paragraph to the text, in the Pose Estimation/Recognition Solutions section, to help readers better understand the potential and limitations of mediapipe: "A clear advantage of MediaPipe is its capability of running on mainstream cellphones (due to its low computational requirements), enabling body tracking applications to be developed aiming these platforms. As a drawback of 3D estimation from 2D content, the Z coordinates of the joints are calculated in relation to a central point of the user, instead of having an absolute value, which happens with conventional RGB-D sensors, that provide the distance of the user to the sensor."
Best regards,
Authors
Reviewer 2 Report
The paper presents an intereseting research work about a comparison among several motion capture systems to understand the quality of the new generation of algorithms for motion tracking. A marker-based mocap system has been used as gold standard and several RGB-D markerless mocap system has been compared with Movenet, which is a AI SDK for human motion tracking. The aim is the measurement of accuracy of movement specifically correlated to the medical field.
If even if the paper is well written and the results are very interesting, the authors forgot to consider the last generation of Microsoft Kinect device, i.e., the Kinect Azure device. This RGB-D device allows a better accuracy thant the old (and out of production) Kinect v2.
Several reserach works are already available about the comparison between Kinect v2 and Kinect Azure.
If we want consider this reserach innovative the authors should mention some reserach work in the scientifi backgound of the paper. Also, the authors should done the well-done validation test considering also the kinect azure device.
A couple of work about the comparison between Kinect v2 and Kinect Azures are in the following list:
- https://doi.org/10.3390/s21020413
- https://doi.org/10.1007/978-3-030-79763-8_43
There are some minore reivew that I would like to suggest at the athors:
- Figure 1: the caption is too long. The authors should describe the workflow in the section.
- Tables: the authors should use the squared brackets for the meaurements units.
-Figure 6: the authos should add the measurement units labels in each row of graphs.
-Figure 7 and Figure 8: labels of measurements unity and increasing the font size of legenda.
The paper can be accepted after a major review.
Author Response
Dear Reviewer,
We really appreciate all the suggestions and comments regarding our paper. We would like to say that we carefully considered all suggestions and items pointed out. We describe how we dealt with each of the points highlighted by the two reviewers as follows.
Reviewer 2:
Reviewer comment: "If even if the paper is well written and the results are very interesting, the authors forgot to consider the last generation of Microsoft Kinect device, i.e., the Kinect Azure device. This RGB-D device allows a better accuracy thant the old (and out of production) Kinect v2.
Several reserach works are already available about the comparison between Kinect v2 and Kinect Azure.
If we want consider this reserach innovative the authors should mention some reserach work in the scientifi backgound of the paper. Also, the authors should done the well-done validation test considering also the kinect azure device.
A couple of work about the comparison between Kinect v2 and Kinect Azures are in the following list:
- https://doi.org/10.3390/s21020413
- https://doi.org/10.1007/978-3-030-79763-8_43"
Author's comment: We have added a second paragraph of method section explaining why kinect azure was not used in this work and mentioned the suggested work as well.
Reviewer comment: Figure 1: the caption is too long. The authors should describe the workflow in the section.
Author's comment: The figure caption was reduced and the rest of the description was added to the section as a new paragraph.
Reviewer comment: Tables: the authors should use the squared brackets for the meaurements units.
Author's comment: Tables were fixed according to the suggestion.
Reviewer comment: Figure 6: the authors should add the measurement units labels in each row of graphs.
Author's comment: Figure was fixed according to the suggestion.
Reviewer comment: Figure 7 and Figure 8: labels of measurements unity and increasing the font size of legenda.
Author's comment: Figures 7 and 8 were fixed according to the suggestions.
Best regards,
Authors
Round 2
Reviewer 2 Report
The authors have modified the paper according to the reviewers' requests. Therefore, the paper can be accepted.
Author Response
Thank you for all the precious comments!